# High-Temperature Effect on the Tensile Mechanical Properties of Unidirectional Carbon Fiber-Reinforced Polymer Plates

**DOI:** 10.3390/ma14237214

**Published:** 2021-11-26

**Authors:** Yongqiang Zhang, Yue Li, Jialei Zhang, Jinwu Pan, Li Zhang, Fuli Tan, Hongjian Wei, Wei Zhang

**Affiliations:** 1Institute of Fluid Physics, China Academy of Engineering Physics, Mianyang 621900, China; minizhang_0804@163.com (Y.Z.); zhangjialei21@126.com (J.Z.); zhangli8037@sina.com (L.Z.); tanfuli2008@163.com (F.T.); 2High Velocity Impact Dynamics Lab, Harbin Institute of Technology, Harbin 150080, China; 21B918039@stu.hit.edu.cn (Y.L.); pjw18285597@163.com (J.P.); weehj123@126.com (H.W.)

**Keywords:** high-temperature resistance, carbon fiber-reinforced polymer (CFRP), tensile mechanical properties, laser transducer system

## Abstract

Carbon fiber-reinforced polymer (CFRP) has the advantages of a high strength-weight ratio and excellent fatigue resistance and has been widely used in aerospace, automotive, civil infrastructure, and other fields. The properties of CFRP materials under high temperatures are a key design issue. This paper presents the quasi-static tensile mechanical properties of unidirectional CFRP plates at temperatures ranging from 20 to 600 °C experimentally. The laser displacement transducer was adopted to capture the in situ displacement of the tested specimen. The results showed that the tensile strength of the CFRP specimen was affected by the high-temperature effect significantly, which dropped 68% and 16% for the 200 and 600 °C, respectively, compared with that of the room temperature. The degradation measured by the laser transducer system was more intensive compared with previous studies. The elastic modulus decreased to about 29% of the room temperature value at 200 °C. With the evaporation of the resin, the failure modes of the CFRP experienced brittle fracture to pullout of the fiber tow. The study provides accurate tensile performance of the CFRP plate under high-temperature exposure, which is helpful for the engineering application.

## 1. Introduction

Carbon fiber-reinforced polymer (CFRP) has the advantages of high strength, light weight, and excellent fatigue resistance and has been widely used in aerospace, automotive, civil infrastructure, and other fields [1,2,3]. The widespread application of CFRP has brought many challenges to engineers. In some special engineering application, the firing resistance and high-temperature resistance of the CFRP plate are significant performance indexes, resulting in the importance of digging the degradation of the mechanical performance of CFRP at high temperatures. One of the challenges is to understand that when CFRP is exposed to a high temperature and harsh environment, its fire resistance is an important issue that needs further research due to the high-temperature softening [4]. CFRP strength and stiffness will be decreased noticeably as the exposure temperature exceeds the glass transition temperature (Tg) at which the polymer resin changes from a glassy state to rubbery-like state [5,6]. It is necessary to perform comprehensively the high-temperature effect on mechanical properties of CFRP materials.

The temperature effect on the fiber-reinforced laminates have been extensively studied to foster understanding of the mechanism of the glass transition temperature. Grape et al. [7] studied the effect of temperature on the compression properties of two-dimensional braided composites, which indicated that the failure mode changed from shear cracking and delamination failure to the plastic KinKing failure as the temperature exceeded the glass transition temperature. The temperature effect was also certified by Dang et al. [8], which had noticeable influence on the bending performance and failure mode of braided composites. Cao et al. [9] studied the effect of temperature on the tensile strength of CFRP with two resins at the temperature between 20 °C and 120 °C. Ashrafi et al. [10] investigated the high-temperature effect on the mechanical properties of spiral-wound CFRP reinforcement and two different resin channel CFRPs. The results showed that the strength of the CFRP decreased to 50% of its initial value in the temperature range of 330 to 450 °C. Zhou et al. [11] studied the mechanical properties of CFRP bars at room temperature to 500 °C. The results showed that the strength decreased sharply when the temperature exceeded the glass transition temperature. Hawileh et al. [12] experimentally studied the variation of tensile strength and elastic modulus of CFRP, GFRP, and their mixed combination (CG) at different temperatures, ranging from 25 to 300 °C. According to the results revealed by Bisby [13], who stated the fire resistance of CFRP at temperature below 450 °C. Hence, a temperature above 450 °C should be discussed to determine the firing resistance of the composite. Wang et al. [14] tested the CFRP plates from 20 to 700 °C. The results showed that a CFRP plate fired at 350–600 °C due to decomposition of resin and oxidation of carbon fiber. The stress-displacement curve was linear below 530 °C, and the strength decreased to 50% of room temperature value at 300 °C. Feih et al. [15] conducted a tensile test on carbon fibers and CFRP laminate under a high temperature of 250–700 °C, and the results indicated that the tensile strength and modulus of carbon fiber were reduced by nearly 50% following exposure to temperatures over the range of 400–700 °C due to the oxidation of the higher stiffness layer in the near-surface fiber region. Based on the high-temperature mechanical properties of laminates and their components, researchers conducted comprehensive studies on the composite sandwich plate exposure to a high-temperature environment [16,17,18,19]. These experimental tests were conducted on a universal test machine, and the data were recorded by the displacement transducer on the machines. The experimental error will not be discussed comprehensively.

To obtain the accurate test results, a lot of new measuring methods were employed with the test machine under high temperatures. Yu and Kodur [20] investigated the tensile strength and elastic modulus degradation of the CFRP strip and CFRP rod in a 20–600 °C temperature range. A linear variable differential transformer (LVDT) sensor was used to measure the strain of the specimen. The results showed that the tensile strength and elastic modulus properties degraded significantly beyond 300 °C due to decomposition of FRP resin. However, the strain of the specimen was not a real strain and included the slip of the bond medium. To study fire resistance, Xu et al. [21] carried out a type of high-temperature-resistant CFRP tendon in the range from 20–600 °C. In order to resolve the problem of measurement in high temperature, the digital image correlation (DIC) technology was adopted to capture the in situ strain on the surface of the CFRP laminates. However, the results showed that the DIC can only be valid at a temperature below 350 °C due to the failure of the spackles on the specimen.

In general, the study on the in situ measurement of the mechanical properties for the CFRP plate at high temperature was limited. In this paper, high-temperature strain gauges and laser displacement transducers were used to measure in situ displacement of the specimen after high-temperature exposure to obtain the strength and modulus. Combining with the in situ strain recorded by the laser transducer, the failure modes and associated mechanisms of the CFRP laminate were discussed.

## 2. Experimental Program

### 2.1. Test Specimens

The unidirectional CFRP plate was composed of SYT55S-12K carbon fiber and epoxy resin (Zhongfu Shenying Carbon Fiber Co., Ltd., Lianyungang, China). Rectangular CFRP plates (400 mm × 400 mm × 2 mm) were fabricated using the hot pressing process with 16 plies of unidirectional carbon fiber prepreg material. The curing process was carried out at a constant pressure of 3 MPa and the temperature conditions were a rising temperature to 80 °C for 60 min, then further rising to 165 °C for 90 min, and finally cooling down to room temperature. Material parameters are shown in Table 1. Table 2 shows the details of unidirectional CFRP plate specimens at room temperature. A total of 30 specimens were tested. According to ASTM D 3039/D 3039M-08 [22], the tensile specimens were cut into the standard specimens, as shown in Figure 1, with a width of 15 mm and a length of 240 mm for each specimen. The GFRP sheet with rough surfaces was used as the clamp tabs, which provided high friction and prevented crushing and sliding of the CFRP in the grip area. The micrometer was used to measure the actual cross-section size of the specimen in the center distance area. This size was used to calculate the strength of the CFRP.

### 2.2. Test Device and Procedure

All tests were carried out on ZWICK-100KN testing machine (Zwick/Roell Testing Technology (Shanghai) Co., Ltd., Shanghai, China). The tensile loads and displacements were recorded by the testing machine. Temperature was applied by a high-temperature electric furnace (Self-design) controlled by a temperature controller. Figure 2 shows a high-temperature tensile device set-up, including a universal testing machine fixture, an installed electric furnace, a laser displacement sensor (KEYENCE (CHINA) CORPORATION, Shanghai, China), a reference metal strip, and a CFRP specimen. A simplified schematic of the high-temperature electric furnace and specimen deformation measurement method details are shown in Figure 3. The electric furnace, with a maximum temperature of 1500 °C and obtained heat from one far-infrared spiral carbon fiber quartz heating tube with a diameter of 70 mm, length of 90 mm, and 1000 W, controlled its heat efficiency by tuning the applied voltage to extend the used life of the heating tube of the electric furnace, realizing uniform heating and temperature control. A thermocouple was used to measure the electric furnace’s chamber temperature. The temperature was the real-time temperature of the specimen. The furnace was not sealed, and the bottom and top were open for oxygen circulation. In order to model the real firing resistance environment, the mechanical behaviors of specimens at 22 °C, 200 °C, 300 °C, 400 °C, 500 °C, and 600 °C were adopted. Before the tensile test, the specimen was heated to the specified temperature and held at that temperature for 20 min to keep the plate section at a constant temperature, thus causing physical and chemical changes in the epoxy resin. Then, a tensile load was applied to the specimen until the specimen failed. During the heating and holding phases, the specimen was clamped at one end. The original unclamped end of the specimen was closed before loading. Then, the specimen was loaded (while still subject to temperature) to failure at 0.75 mm/min in beam displacement control mode.

As shown in Figure 3, the axial strain under a high-temperature tensile test should be measured by high-temperature strain foil (FBAB 120-3BB250(2)-x) (ZEMIC, Xian, China) and laser displacement transducer (LK-H155) (KEYENCE (CHINA)CORPORATION, Shanghai, China). The high-temperature strain foils were used to measure the tensile axial strain under 200 °C, which were stuck on the front and back faces of the specimen at the mid of the gauge length; the laser displacement transducers were used to measure the tensile axial strain within the temperature range of 200–600 °C. Two reference metallic strips were fixed at the two ends of the gauge length of the specimen, as shown in Figure 3. Every laser transducer was paired with one reference metallic strip; so, the displacement of reference metallic strip could be tracked by the laser transducer when the specimen was stretched. The specimen’s tensile axial strain was the displacement difference measured by the two laser transducers.

Figure 4 shows the typical displacement-time curves captured by the testing machine and laser transducers as the CFRP laminate exposure at 200 °C. In the figure, d_b_ and d_t_ represent the displacement of the bottom and the top laser transducers, respectively. The Δd is the difference between the d_b_ and d_t_, which indicates the in situ displacement of the specimen. The dm is the result recorded by the testing machine. It is clear that the error between the two measurements increased with the testing time, with the maximum error exceeding 100%. The result demonstrated that the load cells in the machine can be influenced significantly during the test process, and the laser transducer measurement technology is an ideal measurement method for a CFRP high-temperature test.

## 3. Test Results

### 3.1. Strength and Elastic Modulus

Experimental results of the five repeated tests are listed in Table 3 and Table 4. According to ASTM D 3039/D 3039M-08 [22], the tensile strength and modulus of the tested CFRP laminate were calculated. Taking the data acquired at room temperature as the benchmark, the degradation ratio of the tensile strength and modulus are listed as well. The failure modes after the tests were observed and are categorized in the tables. At room temperature, the crossed strain gauges were used to obtain the Poisson’s ratio, which varied, from 0.33 to 0.17 as temperature increased from 22 to 200 °C, due to the temperature softening of the composite.

The typical tension-displacement curves of the testing machine at different temperatures are shown in Figure 5. The maximum tensile displacement decreased with the increase of temperature: 10.75 mm, 9.15 mm, 8.65 mm, 7.90 mm, 7.30 mm, and 3.65 mm for the temperature of 22 °C, 200 °C, 300 °C, 400 °C, 500 °C, and 600 °C, respectively. Compared with the fracture strain at room temperature, the fracture strains at 200 °C, 300 °C, 400 °C, 500 °C, and 600 °C dropped 14.8%, 19.5%, 26.5%, 32.1%, and 66.0%. The sharp decrease of the fracture strain, especially at 600 °C, was the dominant factor for the different failure modes.

Figure 6 shows the tensile strength and the associated strength retention ratio of the CFRP laminates under the different temperatures. The experimental result conducted by Wang et al. [14] is plotted in the figure to make a comparison with the data captured by the laser transducer system. Due to the different manufacturing procedures and component parameters, the strength retention ratio was used to replace the strength obtained by Wang et al. [14]. The ultimate tensile strength of the CFRP plate exposure to high temperature was focused by the researchers for decades, resulting in several models to predict the tendency. Therefore, the sigmoid function proposed by Bisby [13] was adopted to fit the ultimate strength in Wang et al. [14]. The results showed the CFRP laminate tested in the current paper had a similar tendency with the references. According to the glass transition temperature (T_g_) and evaporation of the resin (T_decomp_), the data captured by the laser transducers exhibited more intensive changes between the T_g_ and T_decomp,_ in which the tensile strength measured by previous studies was supposed to be a plateau. The real tensile strength of the CFRP laminate experienced rapid degradation as the temperature exceeded 400 °C.

Figure 7 shows the variety curve of specimen elastic modulus with temperature. The ratio of elastic modulus at various temperatures to elastic modulus at room temperature is also shown in this figure. During the test, it was found that the furnace emitted white smoke when the temperature was greater than 230 °C, indicating that the resin had decomposed and volatilized. As can be seen from Figure 7, when the temperature exceeded 22 °C, the elastic modulus decreased rapidly. The elastic modulus remained at 29% at 200 °C. This was inconsistent with the results of previous studies [13]. The reason for this result is that the tensile properties of the fiber composites were controlled by the fiber; the fiber oxidation and resin softening had significant effect on the elastic modulus.

### 3.2. Failure Modes

Failure modes of the tensile specimens at different temperatures are shown in Figure 8. Three types of failure mode can be named as Mode I, Mode II, and Mode III.

One sample at 300 °C, 400 °C, 500 °C, and 600 °C was selected, and there were four samples in total. The above four damaged samples were photographed by electron microscope. Scan photos are shown in Figure 9. By contrast, it was found that the resin on the carbon fiber decrease as the increase of temperature.

#### 3.2.1. Mode I

Within the temperature range of 20 °C~200 °C, the resin temperature was below the glass transition temperature and was in an elastic state [5]. The failure mode of the specimen was fiber brittle fracture. The sound of some fibers breaking could be heard before the specimen was destroyed. At the moment when the specimen was damaged, this sound was accompanied by a snapping sound.

#### 3.2.2. Mode II

For the temperature range of 200~400 °C, the resin softened significantly, and the strength decreased slightly more slowly. The carrying capacity of the specimen was reduced after resin softening and fibers’ fracture. There were still fibers connected with resin, and some of them formed hardened bundles of fibers as they cooled. Fibers and fiber bundles’ disorder were the main features of this failure mode.

#### 3.2.3. Mode III

For the temperature range of 400~600 °C, the specimen failed in Mode III and its strength decreased rapidly. When the temperature reached 400 °C, the resin evaporated enough to leave a soft bundle of fibers. The bundle was oxidized, resulting in a loss of tensile strength and modulus.

## 4. Summary and Conclusions

In this paper, the mechanical properties of the CFRP laminate were tested at 22 °C, 200 °C, 300 °C, 400 °C, 500 °C, and 600 °C. The laser transducer system was adopted to provide more accurate experimental data under high temperatures and avoid the experimental error caused by the testing machine. Combining with the failure modes, the tensile strength and modulus of the CFRP laminate exposure to 20–600 °C were analyzed.

The elastic modulus and tensile strength of a unidirectional CFRP plate decreased at high temperature significantly, exhibiting a more rapid decrease compared with the proposed model and previous results. At 200 °C, the tensile strength of the unidirectional CFRP plate specimen decreased to about 68% of the room temperature value and the elastic modulus decreased to about 29% of the room temperature value.

In the temperature range of 20 °C to 200 °C, the failure mode of the specimen was similar to that of the room temperature specimen. The fiber brittle fracture occurred along the specimen gauge length. In the temperature range of 200 °C to 400 °C, the epoxy adhesive softened and the specimen failure was mainly due to the partial loss of the epoxy adhesive and the cracking of the plate. At 400 °C, epoxy adhesive was burned and fiber fracture led to specimen failure.

## Figures and Tables

**Figure 1 materials-14-07214-f001:**
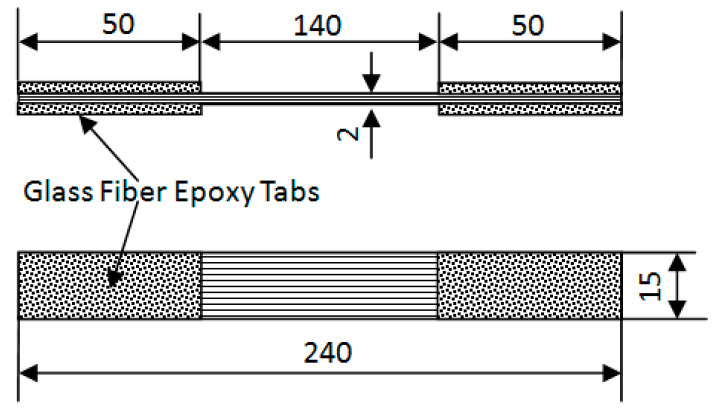
Schematic of tensile specimen geometry (unit: mm).

**Figure 2 materials-14-07214-f002:**
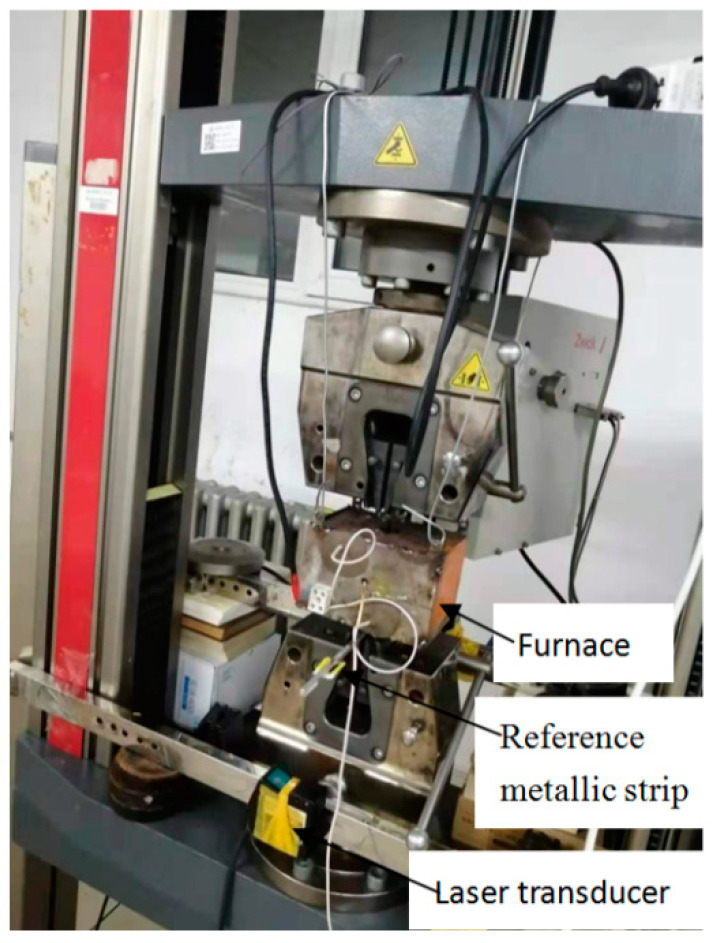
Test setup.

**Figure 3 materials-14-07214-f003:**
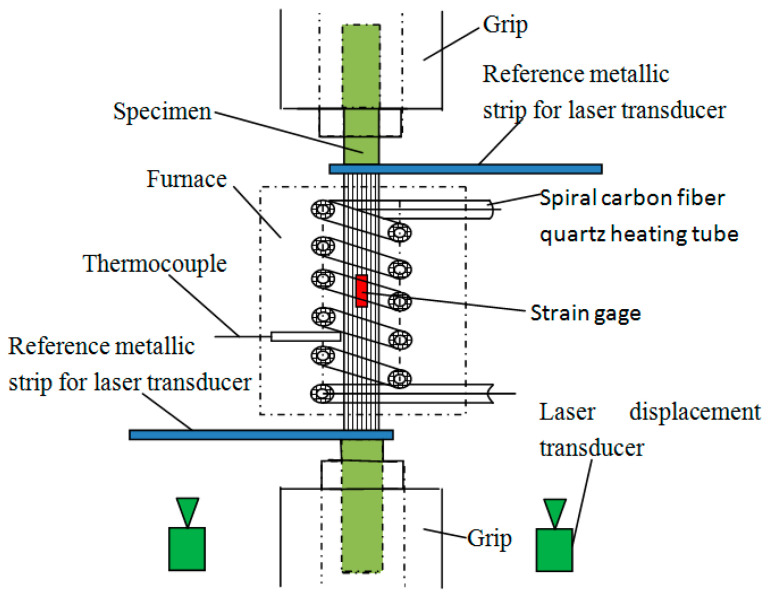
High-temperature electric furnace and specimen deformation measurement method details.

**Figure 4 materials-14-07214-f004:**
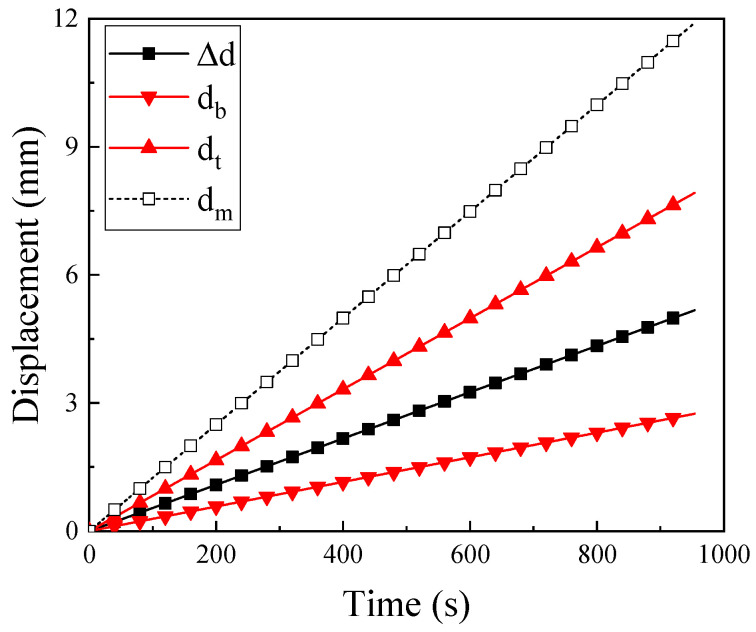
The typical results of displacement of testing machine and laser transducers.

**Figure 5 materials-14-07214-f005:**
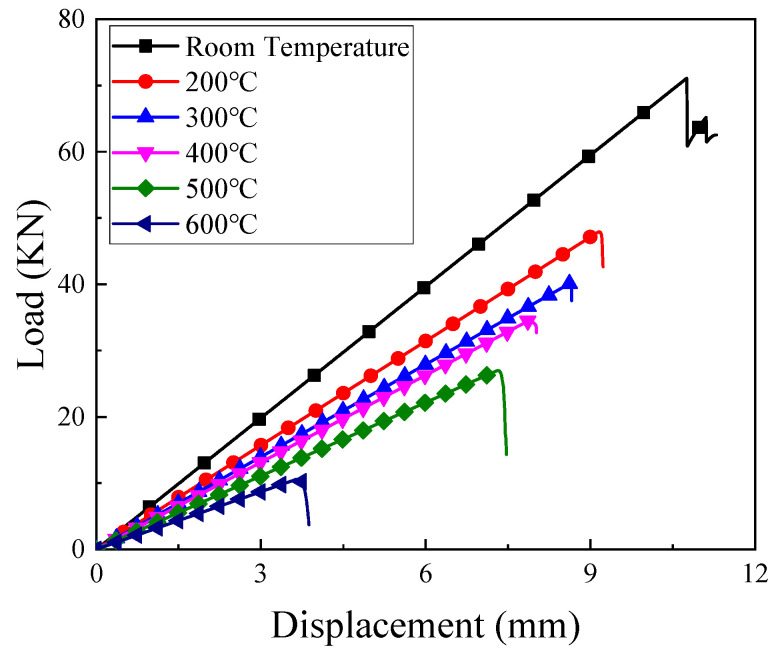
Typical test machine tensile force versus displacement curves at different temperatures.

**Figure 6 materials-14-07214-f006:**
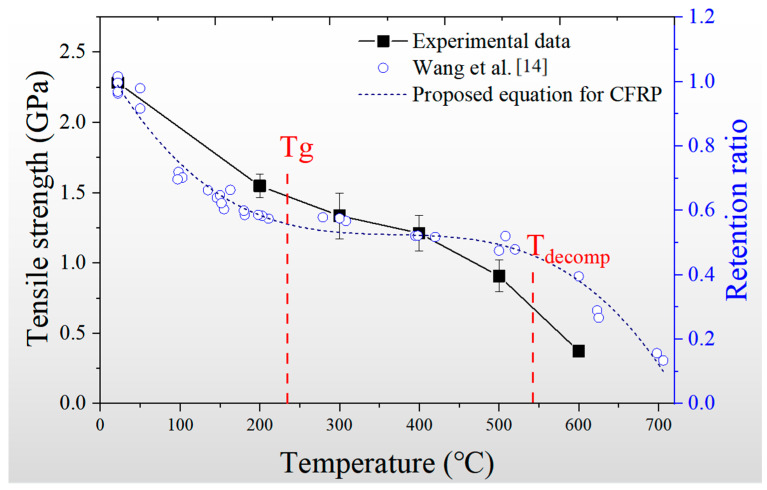
Variety curve of strength with temperature.

**Figure 7 materials-14-07214-f007:**
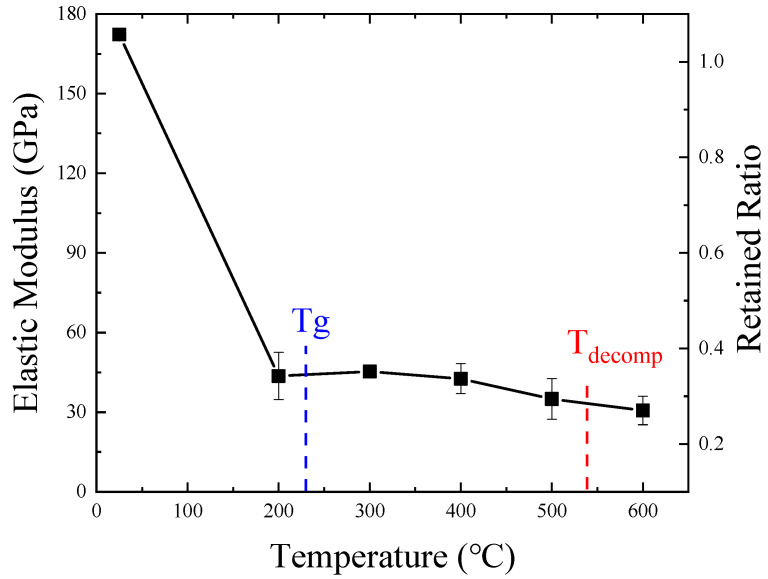
Variety curve of elastic modulus with temperature.

**Figure 8 materials-14-07214-f008:**
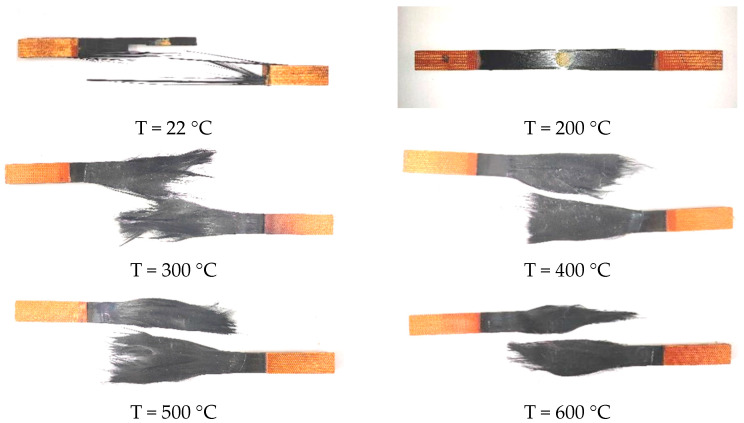
Failure modes of CFRP plate specimens at different temperatures.

**Figure 9 materials-14-07214-f009:**
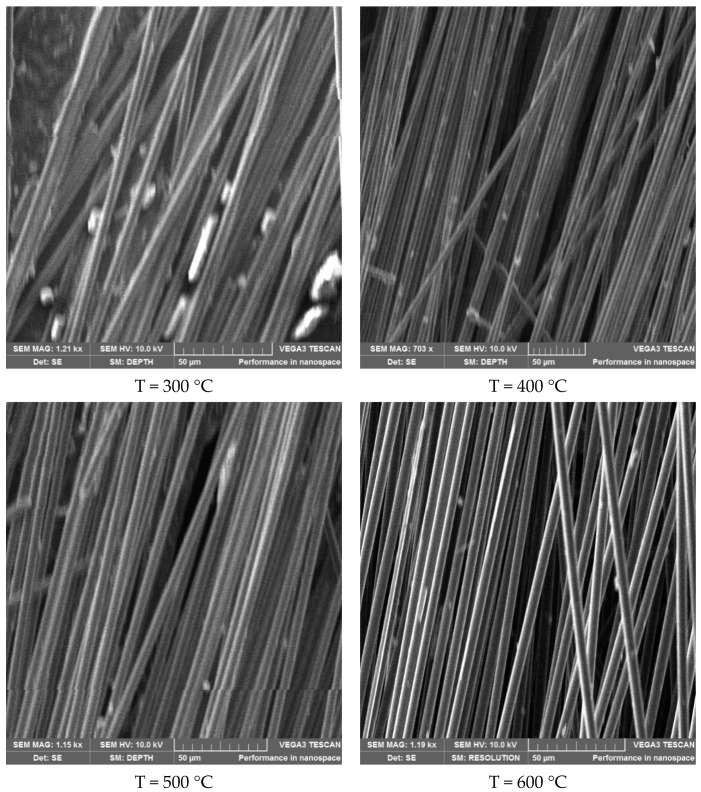
Sem images of CFRP plate specimens at different temperatures.

**Table 1 materials-14-07214-t001:** Properties of component materials for unidirectional CFRP plate specimens.

Material	Type	Tensile Strength (MPa)	Tensile Modulus (GPa)	Elongation (%)	Density (g/cm^3^)	Tg (°C)
Carbon fiber	SYT55S-12K	5900	295	1.9	1.79	-
Resin	BA 202	80	3.5	2.9–3.8	1.17	230–235

**Table 2 materials-14-07214-t002:** Properties of unidirectional CFRP plate specimens at room temperature.

Thickness (mm)	Fiber Content by Volume %	Tensile Strength (MPa)		Tensile Modulus (GPa)		Fracture Strain %
		Mean	Min	Mean	Min	
2.0	68 ± 2	2300	2227	172	171	1.7

**Table 3 materials-14-07214-t003:** Test results of the tensile strength of unidirectional CFRP plate specimens.

No.	Temperature (°C)	Tensile Strength (MPa)	Poisson’s Ratio	Strain (uε)	Average Strength (MPa)	Retention Ratio	Failure Mode
T1-1	22	2385.596	0.317	15,775	2281.71	1	I
T1-2		2227.756	0.330	17,516			I
T1-3		2280.480	0.332	15,735			I
T1-4		2268.118	0.332	15,772			I
T1-5		2246.586	0.329	16,498			I
T2-1	200	1552.713	0.187	35,112	1547.58	0.68	I
T2-2		1467.448	0.183	6638			I
T2-3		1461.829	0.140	6391			I
T2-4		1657.283	0.176	28,571			I
T2-5		1598.628	0.179	30,683			I
T3-1	300	1442.481	--	32,920	1335.03	0.59	II
T3-2		1487.260	--	31,923			II
T3-3		1087.198	--	46,630			II
T3-4		1401.766	--	38,720			II
T3-5		1256.426	--	35,209			II
T4-1	400	1191.488	--	33,157	1211.56	0.53	III
T4-2		1323.174	--	28,538			III
T4-3		1052.650	--	26,440			III
T4-4		1138.193	--	27,230			III
T4-5		1352.295	--	27,061			III
T5-1	500	1042.207	--	19,660	906.32	0.40	III
T5-2		992.776	--	30,820			III
T5-3		765.653	--	24,480			III
T5-4		834.294	--	20,896			III
T5-5		896.661	--	22,916			III
T6-1	600	409.165	--	10,630	371.95	0.16	III
T6-2		367.555	--	13,018			III
T6-3		317.591	--	13,307			III
T6-4		378.910	--	12,040			III
T6-5		386.550	--	12,461			III

**Table 4 materials-14-07214-t004:** Test results of the tensile elastic modulus of unidirectional CFRP plate specimens.

No.	Temperature (°C)	Tensile Modulus (GPa)	Average Modulus (GPa)	Retention Ratio
T1-1	22	173.631	171.16	1
T1-2		171.531		
T1-3		171.303		
T1-4		169.083		
T1-5		170.289		
T2-1	200	44.221	50.48	0.29
T2-2		51.303		
T2-3		47.947		
T2-4		55.685		
T2-5		53.293		
T3-1	300	50.379	38.88	0.23
T3-2		46.588		
T3-3		23.315		
T3-4		36.202		
T3-5		37.958		
T4-1	400	35.894	42.59	0.25
T4-2		46.365		
T4-3		38.944		
T4-4		41.798		
T4-5		49.975		
T5-1	500	46.365	36.74	0.21
T5-2		32.212		
T5-3		31.276		
T5-4		37.549		
T5-5		36.318		
T6-1	600	38.521	30.61	0.18
T6-2		28.233		
T6-3		23.878		
T6-4		31.402		
T6-5		31.028		

## Data Availability

The data presented in this study are available on request from the corresponding author.

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
