# Peer review of "High-Temperature Effect on the Tensile Mechanical Properties of Unidirectional Carbon Fiber-Reinforced Polymer Plates"

_materials, 2021, doi:10.3390/ma14237214_

Round 1

Reviewer 1 Report

Masuscript entitled: "High temperature effect on the tensile mechanical properties of unidirectional carbon fiber reinforced polymer plates"

The manuscript describes the effect of temperature on the deterioration of tensile strength and elastic modulus of unidirectional carbon fiber reinforced polymer (CFRP) plates. Manuscript written correctly, although it describes the known problem of significant deterioration of mechanical properties during tensile tests for CFRP composites. As polymer matrix an epoxy resin system was applied.

Samples were tested at temperatures raised from 22 to 600 ° C to determine tensile strength and elastic modulus. Authors applied universal mechanical testing system of the specimen with a high-temperature electric quartz tube furnace and applied two reference metallic strips fixed at the two ends of the gauge length of the specimen with possibility of tracking reference metallic by the laser transducer when specimen was stretched.

In my opinion this article is rather the test report than the scientific overview of results.

Starting with the abstract, authors present the effect of temperature conditions on the percentage deterioration of the strength parameters of CFRPs. However, they do not explain any principal purpose of this research. Therefore, by putting the present work in the context of similar studies, it is not clear to me that this work contributes something new. As a result, I think the manuscript cannot be accepted for publication in the present form. Authors needs to further explain the uniqueness and innovation of their research and the present methods in detail. I propose supplement with a summary table presenting comparison of results obtained by authors with the literature in the introduction or during discussion.

The recommendation for this manuscript is major revision.

Below some comments:

Wrong wording can be found in Introduction, exactly in the fragment: "The resin changes from glass to rubberization if the temperature large than the glass transition temperature (Tg), CFRP strength and stiffness will be decreased [6-7].". The correct version should be "from glass to rubber-like material" or "from glassy state to rubbery-like state".

 It should also be mentioned that an appropriately selected epoxy matrix of CFRPs allows to significantly shift the Tg up to approx. 300° C. Authors should give information about type of epoxy resin and curing agent, because it determines the thermal and mechanical properties of epoxy binder, and finally CFRP plates. The trade name of the resin BA 202 produced by Zhongfu Shenying Carbon Fiber Co given in Table 1 is not enough. During the review of the previous version of the article, I also asked for it.

In discussion concerning Section 3.2.3. authors conclude "when the temperature reaches 400°C, the resin evaporates enough to leave a soft bundle of fibers."

To draw conclusions about the mass loss of resin in the composite, it is worth to perform detailed thermogravimetric analysis to determine the loss of mass as a function of temperature. On what basis do the authors conclude that the resin enough evaporates form CFRP at 400° C?

Author Response

Response to Reviewer 1

The manuscript describes the effect of temperature on the deterioration of tensile strength and elastic modulus of unidirectional carbon fiber reinforced polymer (CFRP) plates. Manuscript written correctly, although it describes the known problem of significant deterioration of mechanical properties during tensile tests for CFRP composites. As polymer matrix an epoxy resin system was applied.

Samples were tested at temperatures raised from 22 to 600 ℃ to determine tensile strength and elastic modulus. Authors applied universal mechanical testing system of the specimen with a high-temperature electric quartz tube furnace and applied two reference metallic strips fixed at the two ends of the gauge length of the specimen with possibility of tracking reference metallic by the laser transducer when specimen was stretched.

  1. In my opinion this article is rather the test report than the scientific overview of results.

Response:  We appreciate the reviewer’s valuable suggestions. The present article was totally an experimental study on the heated CFRP plates, which aimed to provide accurate measurement of the in-situ displacement with the laser transducer. According to the reviewer, the associated analysis of the mechanisms of the failure modes were added to the revision.

  1. Starting with the abstract, authors present the effect of temperature conditions on the percentage deterioration of the strength parameters of CFRPs. However, they do not explain any principal purpose of this research. Therefore, by putting the present work in the context of similar studies, it is not clear to me that this work contributes something new. As a result, I think the manuscript cannot be accepted for publication in the present form. Authors needs to further explain the uniqueness and innovation of their research and the present methods in detail. I propose supplement with a summary table presenting comparison of results obtained by authors with the literature in the introduction or during discussion.

Response:  Thanks for the comments. As mentioned in the above response, the main purpose of the article was to provide accurate in-situ measurement of the displacement under the high temperature. The laser transducer made the in-situ measurement possible. Based on the experimental results, we aimed to provide valuable suggestions to the engineering application. Hence, we rewritten the abstract and some other parts to highlight the principal purpose and the value of the current study in the revision.

As for the comparison with the previous literatures, we added some introductions of the similar literatures in the revised introduction. However, the material system of the CFRP/Epoxy was SYT55S -12K fiber and BA 202 epoxy polymer, which quite different from the previous studies that considering the mechanical properties under high temperatures. Hence, the comparison of the experimental results between the article and previous studies had not be added into the current manuscript.

  1. Wrong wording can be found in Introduction, exactly in the fragment: "The resin changes from glass to rubberization if the temperature large than the glass transition temperature (Tg), CFRP strength and stiffness will be decreased [6-7].". The correct version should be "from glass to rubber-like material" or "from glassy state to rubbery-like state".

Response:  Thanks for the carefully review on the manuscript. According to the reviewer, we modified the wrong terms throughout the manuscript.

  1. It should also be mentioned that an appropriately selected epoxy matrix of CFRPs allows to significantly shift the Tg up to approx. 300° C. Authors should give information about type of epoxy resin and curing agent, because it determines the thermal and mechanical properties of epoxy binder, and finally CFRP plates. The trade name of the resin BA 202 produced by Zhongfu Shenying Carbon Fiber Co given in Table 1 is not enough. During the review of the previous version of the article, I also asked for it.

Response:  Thanks for the suggestion. The CFRP/Epoxy laminate used in the study was manufactured from the CFRP/Epoxy prepreg by the hot pressing. With the given material properties in the previous version, we added the stacking sequence of the laminate in the revision. With the properties of the components and laminate, we believe that the information of the material is enough, as the same with the most literatures.

It is indeed that the Tg of some CFRP/Epoxy prepregs is up to approx. 300° C. But there are some other CFRP/Epoxy prepregs exhibiting low Tg. For example, Zhou et al.[1] and Feih et al. [2]  mentioned in their paper that the Tg of the CFRP/Epoxy prepreg is about 140° C and 128,respectively. We thought the Tg of the given epoxy, about 230° C, was reasonable.

[1] Junmeng Zhou, Yu Wang, et al., Temperature effects on the compressive properties and failure mechanisms of composite sandwich panel with Y-shaped cores, Composites Part A 114 (2018) 72–85.

[2] Feih S, Mouritz A P. Tensile properties of carbon fibres and carbon fibre–polymer composites in fire[J]. Composites Part A: Applied Science and Manufacturing. 2012, 43(5): 765-772.

  1. In discussion concerning Section 3.2.3. authors conclude "when the temperature reaches 400°C, the resin evaporates enough to leave a soft bundle of fibers."

To draw conclusions about the mass loss of resin in the composite, it is worth to perform detailed thermogravimetric analysis to determine the loss of mass as a function of temperature. On what basis do the authors conclude that the resin enough evaporates form CFRP at 400° C?

Response:  Thanks for the valuable suggestion. In the revision, we added some SEM images of the heated CFRP laminate after the test. It is clear that there was no epoxy resin among the fibers, which validated the conclusion we mentioned in the paper.

Reviewer 2 Report

The manuscript features an introduction, review of the literature, methods, and test results. The abstract is concise and well written. The manuscript provides the necessary background information covering enough references. The research methodology is appropriate, but some suggestions are recommended.

1-In the “Introduction” section, it is suggested to cite and discuss more literature relevant to topics of the present research work.

2-About the experimental tests, how many repetitions were conducted to obviate the measuring errors, the repeatability / consistency was checked? And about the equipment’s calibration?

3-In my, opinion, it will be better if in Figure 8 appearing the failure mode close to the CFRP plate in study.

4-More discussion and explanation, about the chose temperature range, in each failure mode should be clarified.

5-If it is possible, authors should increase their discussion with previous research and highlight how their study is providing a different approach to what has been done.

6-In the conclusions, some future directions and research gaps need to be included.

Author Response

The manuscript features an introduction, review of the literature, methods, and test results. The abstract is concise and well written. The manuscript provides the necessary background information covering enough references. The research methodology is appropriate, but some suggestions are recommended.

  • In the “Introduction” section, it is suggested to cite and discuss more literature relevant to topics of the present research work.

Response:  Thanks for the valuable suggestion. According to the reviewer, we modified the Introduction by adding some more relative literature in the revision.

  • About the experimental tests, how many repetitions were conducted to obviate the measuring errors, the repeatability/consistency was checked? And about the equipment’s calibration?

Response:  Thanks for the question. We conducted 5 repeat tests on each temperature to obtain the average strength and modulus of the CFRP laminate under different temperatures.

As for the calibration, we actually submitted another article about the laser measuring system, including the designing, working principle, calibration, and testing. The accuracy of the laser measuring system is far more reliable than the testing machine. Given the article have not been accepted, we did not put the relative information in the manuscript.

  • In my, opinion, it will be better if in Figure 8 appearing the failure mode close to the CFRP plate in study.

Response:  Thanks for the carefully review on the manuscript. We modified the pictures in Fig.8 to exhibit more clear information about the failure modes of the CFRP laminate.

  • More discussion and explanation, about the chose temperature range, in each failure mode should be clarified.

Response:  Thanks for the valuable suggestion. The temperature rang selected in the paper was the real engineering environment for the CFRP laminate. In that case, the firing resistance and high temperature resistance are the significant requirement for the CFRP laminate. In the revision, we added the necessary information.

  • If it is possible, authors should increase their discussion with previous research and highlight how their study is providing a different approach to what has been done.

Response:  Thanks for the valuable suggestion. As for the comparison with the previous literatures, we added some introductions of the similar literatures in the revised introduction. However, the material system of the CFRP/Epoxy was SYT55S -12K fiber and BA 202 epoxy polymer, which quite different from the previous studies that considering the mechanical properties under high temperatures. Hence, the comparison of the experimental results between the article and previous studies had not be added into the current manuscript.

  • In the conclusions, some future directions and research gaps need to be included.

Response:  Thanks for the suggestion. We added the relative information in the revision.

Round 2

Reviewer 1 Report

After reading the authors' detailed responses and corrections to my suggestions I agree to publish their articles in its current form.

This manuscript is a resubmission of an earlier submission. The following is a list of the peer review reports and author responses from that submission.

Round 1

Reviewer 1 Report

The manuscript describes the deterioration of tensile strength and elastic modulus of unidirectional CFRP plates tested at temperatures raised from approximately 22 to 600 ° C. Tensile tests showed that deformation test system failed at 350 ° C because the resin burnt out.To deal with handicaps authors applied two reference metallic strips fixed at the two ends of the gauge length of the specimen with possibility of tracking reference metallic by the laser transducer when specimen is stretched

The recommendation for this manuscript is minor revision.

Below some comments:

In Experimental Part, Table 1 contains the characteristic parameters of epoxy resin commercially known as BA 202 produced by Zhongfu Shenying Carbon Fiber Co. I suggest to put more complex information about curing conditions and what type of cross-linked agent was used.

The Tg of cross-linked epoxy resin strongly dependent on the curing agent structure used and curing conditions. For various curing agents the Tg of epoxy matrix can be differ significantly. So it will have positive effect to give in details such information.

Line 126 in Experimental part

Fragment "to examine the degradation of mechanical properties of CFRP plates under elevated temperatures"

Degradation of mechanical properties?? It could be degradation of structure but deterioration of mechanical properties.

Author Response

Responds to the Editor and Reviewer 1

Dear Editor and Reviewer 1:

Thank you for your letter and for the reviewers’ careful and meticulous reading of the paper entitled “Temperature effect on the tensile mechanical properties of unidirectional carbon-fiber-reinforced polymer (CFRP) plates”. (materials-1387195). The reviews are detailed and helpful to finalize the manuscript. The authors would like to kindly acknowledge them. Here follow our comments to the major concerns and answers to specific points. Reviewers’ comments are repeated in italics and our responses inserted after each comment.

Responds to Reviewer 1

Comment 1: In Experimental Part, Table 1 contains the characteristic parameters of epoxy resin commercially known as BA 202 produced by Zhongfu Shenying Carbon Fiber Co. I suggest to put more complex information about curing conditions and what type of cross-linked agent was used.

Response: We thank the reviewer’s valuable suggestion. Information about curing conditions and type of cross-linked agent was added in our revised manuscript.

Comment 2: The Tg of cross-linked epoxy resin strongly dependent on the curing agent structure used and curing conditions. For various curing agents the Tg of epoxy matrix can be differ significantly. So it will have positive effect to give in details such information.

Response: Thanks for the carefulness of the reviewer. In the revised manuscript, we explained the structure of curing agent and curing conditions of epoxy resin in detail according to the suggestions of reviewers.

Comment 3: Line 126 in Experimental part.Fragment "to examine the degradation of mechanical properties of CFRP plates under elevated temperatures".Degradation of mechanical properties?? It could be degradation of structure but deterioration of mechanical properties.

Response: We thank the reviewer’s valuable suggestion. It is really true as reviewer suggested that it could be degradation of structure but deterioration of mechanical properties.We have made modifications according to the comments of reviewers in our revised manuscript.

Again, we appreciate all of your insightful comments. We worked hard to be responsive to them. Thank you for taking the time and energy to help us improve the paper.

Reviewer 2 Report

The manuscript reports on how increasing service temperatures would affect tensile properties of carbon fibers-reinforced composites (CFRPs). The presented research has a practical importance to deem it of some interest, especially filling a somewhat scarce inventory of CFRP properties at elevated temperatures.

However, the research question that the manuscript is attempting to answer lacks credibility. While presenting the stiffness and strength change with temperature might be of some value, it will not be valuable beyond the specific CFRP investigated. Inference from the studied set of composites would not allow interpretation of general trends.

More importantly, most composites are traditionally used below their glass transition temperature (Tg), and the investigated temperature go much higher than the reported 230~235â—‹C. Only one temperature is investigated below Tg.

Furthermore, the authors fail to include their readers in the rationale for selecting the set of temperature used. Is there a practical significance of those temperatures? Is the composite intended to be used at any of them? It would be much more meaningful to the research community to explore how the mechanical properties change within the useful range, or at least understand why the explored range went to 600â—‹C.

Another fundamental flaw in this study has to do with the tensile test used. The authors state: “The tensile specimens were cut into standard specimens as shown 126 in Fig.1, each specimen has a width of 15 mm and a length of 240 mm, in accordance and 127 compliance with ASTM D 3039/D 3039M-08 [20].” (line 128, manuscript). However, the authors fail to follow the standard. ASTM standard D 3039 states explicitly in section 8 (Sampling and Test Specimen) to “test at least five specimen per test condition.” In fact, all testing standards for that matter indicate that five samples are the least number of test s acceptable. The reason is that below five valid results per test condition, no statistical analysis is possible. The average and standard deviation lose their meaning, and there is no confidence in the results. With this in mind, the results shown in Tables 3 and 4 lose their significance and become anecdotal evidence. The authors are encouraged to follow the ASTM standard they are claiming to follow, and include, at least, five valid results per investigated case. Actually, another outlier analysis should be implemented to ensure the validity of the presented results. This means that the authors must have at least five data points per case after removing the outliers. Otherwise, the presented numbers do not carry any weight.

Another aspect that can be improved is in the materials and methods section, the information included does not include enough information about the investigated composite. What epoxy was used? Ow was it manufactured? The authors indicate:  “A total of 24 specimens were prepared and tested in this experimental 124 investigation to examine the degradation of mechanical properties of CFRP plates under 125 elevated temperatures.” (Line 124-125, manuscript). More details are expected by the readers to understand how the investigated CFRPs were obtained.

Moreover, the properties provided by the supplier and presented in Table 1 need to have the confidence interval or standard deviation. Also, are the numbers presented in Table 2 provided by the supplier as well, obtained experimentally (describe how), or inferred theoretically (explain how), or a mix of these methods? What are the ranges of each property? Besides Tg, all other numbers were provided as single values. Was that a single experimental data? Or was that an average? If so provide how each one was obtained and the confidence interval or standard deviation.

Also, it would be in the best interest of the authors, their readers and research community as a whole, to include a comparison of the obtained results in this study and those of the literature.

Finally, the overall English language in the manuscript requires extensive reviewing to allow the readers to follow ideas discussed.

The work presented in this manuscript, in its current form, does not meet the publication standards, and should be rejected.

Author Response

Responds to the Editor and Reviewer 2

Dear Editor and Reviewer 2:

Thank you for your letter and for the reviewers’ careful and meticulous reading of the paper entitled “Temperature effect on the tensile mechanical properties of unidirectional carbon-fiber-reinforced polymer (CFRP) plates”. (materials-1387195). The reviews are detailed and helpful to finalize the manuscript. The authors would like to kindly acknowledge them. Here follow our comments to the major concerns and answers to specific points. Reviewers’ comments are repeated in italics and our responses inserted after each comment.

Responds to Reviewer #2

General Comments: The work presented in this manuscript, in its current form, does not meet the publication standards, and should be rejected.

Response: The authors are grateful to the referee for careful reading of the paper and valuable suggestions and comments. Below we provide our responses point by point, and modify the manuscript accordingly.

Comment 1: The manuscript reports on how increasing service temperatures would affect tensile properties of carbon fibers-reinforced composites (CFRPs). The presented research has a practical importance to deem it of some interest, especially filling a somewhat scarce inventory of CFRP properties at elevated temperatures. However, the research question that the manuscript is attempting to answer lacks credibility. While presenting the stiffness and strength change with temperature might be of some value, it will not be valuable beyond the specific CFRP investigated. Inference from the studied set of composites would not allow interpretation of general trends.

Response: The authors are grateful to the referee’s useful comments. In the conclusion part of our revised manuscript, the conclusion of this paper is limited to the material studied in this paper. The applicability of the conclusions has not been expanded.

Comment 2: .More importantly, most composites are traditionally used below their glass transition temperature (Tg), and the investigated temperature go much higher than the reported 230~235℃. Only one temperature is investigated below Tg. Furthermore, the authors fail to include their readers in the rationale for selecting the set of temperature used. Is there a practical significance of those temperatures? Is the composite intended to be used at any of them? It would be much more meaningful to the research community to explore how the mechanical properties change within the useful range, or at least understand why the explored range went to 600℃.

Response: Thanks for the carefulness and meticulous of the reviewer. We are so sorry that the research temperature range was not clearly stated in the research background. In the revision, the explored temperature range have been explained. In the present study, the temperature effects of mechanical properties of materials studied in this paper are based on the background of structures under fire conditions which the temperature of the fire can reach more than 1000℃.

Comment 3: Another fundamental flaw in this study has to do with the tensile test used. The authors state: The tensile specimens were cut into standard specimens as shown 126 in Fig.1, each specimen has a width of 15 mm and a length of 240 mm, in accordance and 127 compliance with ASTM D 3039/D 3039M-08 [20]. (line 128, manuscript). However, the authors fail to follow the standard. ASTM standard D 3039 states explicitly in section 8 (Sampling and Test Specimen) to test at least five specimen per test condition. In fact, all testing standards for that matter indicate that five samples are the least number of test s acceptable. The reason is that below five valid results per test condition, no statistical analysis is possible. The average and standard deviation lose their meaning, and there is no confidence in the results. With this in mind, the results shown in Tables 3 and 4 lose their significance and become anecdotal evidence. The authors are encouraged to follow the ASTM standard they are claiming to follow, and include, at least, five valid results per investigated case. Actually, another outlier analysis should be implemented to ensure the validity of the presented results. This means that the authors must have at least five data points per case after removing the outliers. Otherwise, the presented numbers do not carry any weight.

Response: Many thanks for the reviewer’s suggestion. We are very sorry for our negligence of ASTM standards for the number of material samples required.In our revised manuscript, we added tests according to ASTM standards to achieve at least five valid results per investigated case.Moreover, outlier analysis have been implemented to ensure the validity of the presented results.

Comment 4: Another aspect that can be improved is in the materials and methods section, the information included does not include enough information about the investigated composite. What epoxy was used? Ow was it manufactured? The authors indicate:A total of 24 specimens were prepared and tested in this experimental 124 investigation to examine the degradation of mechanical properties of CFRP plates under 125 elevated temperatures. (Line 124-125, manuscript). More details are expected by the readers to understand how the investigated CFRPs were obtained.

Response: Thank you for this great suggestion. In the revision, the details of the epoxy have been added .The manufacturing process of CFRPs is introduced in detail to understand how the investigated CFRPs were obtained.

Comment 5: Moreover, the properties provided by the supplier and presented in Table 1 need to have the confidence interval or standard deviation. Also, are the numbers presented in Table 2 provided by the supplier as well, obtained experimentally (describe how), or inferred theoretically (explain how), or a mix of these methods? What are the ranges of each property? Besides Tg, all other numbers were provided as single values. Was that a single experimental data? Or was that an average? If so provide how each one was obtained and the confidence interval or standard deviation.

Response: We thank the reviewer’s valuable advice. The standard deviation have been added in Table 1 and Table 2 in our revised manuscript. In addition, the acquisition method of each data in Table 2 is described in detail.

Comment 6: Also, it would be in the best interest of the authors, their readers and research community as a whole, to include a comparison of the obtained results in this study and those of the literature.

Response: Thank you for your constructive comments. According to the suggestions of Reviewer, a comparative analysis of the research results of this paper and the literature is added in the revised manuscript.

Comment 7: Finally, the overall English language in the manuscript requires extensive reviewing to allow the readers to follow ideas discussed.

Response: We are grateful to the reviewer’s comments. English has been improved in our revised manuscript.

We tried our best to improve the manuscript and made some changes in the manuscript. These changes will not influence the content and framework of the paper. And here we did not list the changes but marked in red in revised paper.

We appreciate for Editors/Reviewers’ warm work earnestly, and hope that the correction will meet with approval.

Once again, thank you very much for your comments and suggestions.